# Token-Level Early Fusion Model Bridging Text and 3D Electron Density Grids in Chemistry

## Abstract

We introduce 3DGrid-LLM, an early-fusion multimodal foundation model that integrates natural language with 3D electron density grids for molecular and materials science. The model extends a large decoder-only language model with discrete volumetric tokens from a 3D VQGAN, enabling unified token-level processing of spatial and textual information. Trained on diverse datasets, 3DGrid-LLM supports bidirectional generation, multimodal question answering, and retrieval-augmented 3D grid generation. Experiments show consistent improvements over baselines in multimodal VQA, semantic text generation, and property-aligned retrieval, demonstrating accurate and physically consistent outputs. This work establishes a scalable framework for incorporating physically grounded volumetric data into language models.

## 1 Introduction

Understanding the structure–property relationships of molecules and materials remains a fundamental challenge in computational chemistry and materials science Morgan & Jacobs (2020); Takeda et al. (2023a); Jain (2024). Central to this problem is the electron density—a three-dimensional (3D) spatial function that encodes both the geometric configuration and electronic structure of a system Koch et al. (2024); Lee & Kim (2024). Electron density grids, whether obtained from ab initio simulations such as density functional theory (DFT) or reconstructed from crystallographic sources (e.g., CIF files), offer a physically grounded and information-rich representation Kirkpatrick et al. (2021); Marzari et al. (2021); Kelley et al. (2024). However, their potential remains largely underexploited in machine learning pipelines for molecular and materials modeling Unke et al. (2021); Fiedler et al. (2022).

Despite recent advances in deep learning for molecules and materials, most approaches rely on 1D or 2D representations such as SMILES strings Soares et al. (2025b); Pan (2023); Ross et al. (2022), graphs Guo et al. (2022); Liu et al. (2023); Takeda et al. (2023b), or engineered descriptors Soares et al. (2023; 2025a); Zhao et al. (2023), which often omit detailed 3D information. Methods that incorporate structure typically do so through atomistic point clouds or geometric graphs Schütt et al. (2021); Poulenard & Guibas (2021); Fang et al. (2022), abstractions that operate at the atomic level and struggle to capture the fine-grained spatial and electronic features encoded in the full density distribution. Moreover, many existing models are optimized for narrow tasks or domains, limiting their ability to generalize across applications Bommasani et al. (2021).

Recent multimodal foundation models in chemistry have begun to address these limitations Choi et al. (2025); Zhou et al. (2023). However, most adopt late fusion architectures, processing each modality independently with dedicated encoders or decoders before combining them at a later stage Soares et al. (2024); Livne et al. (2024); Priyadarsini et al.. This separation can limit the model's capacity to learn joint representations and capture interactions between spatial (e.g., 3D structure) and textual (e.g., scientific language) modalities. In this work, we introduce 3DGrid-LLM, a family of early-fusion multimodal foundation models capable of bidirectional generation and reasoning over scientific text and 3D electron density grids. These grids, derived from small molecules or inorganic materials, are tokenized using a 3D-VQGAN Soares et al. (2025c). The model accepts fused input sequences of grid tokens and language prompts, and supports both 3D-to-text (e.g., property description) and text-to-3D (e.g., density grid generation and retrieval) tasks. This unified framework

enables downstream applications such as scientific question answering, grid-based retrieval, and inverse design.

Extensive evaluations demonstrate that 3DGrid-LLM performs effectively across a diverse set of tasks. We evaluate the model on multimodal visual question answering (VQA), text generation, and grid-based retrieval benchmarks. More importantly, 3DGrid-LLM enables novel capabilities not supported by prior models, including bidirectional generation and multimodal reasoning over scientific text and 3D electron density grids. This flexibility positions 3DGrid-LLM as a unified interface for both interpretability and generation tasks across molecular and materials science domains.

## 2 OVERVIEW OF THE PROPOSED APPROACH

This section outlines the core methodology behind 3DGrid-LLM, highlighting its architecture, pre-training datasets, training pipeline, and generative capabilities. Figure 4 illustrates the general schema for pre-training and multimodal generation of 3DGrid-LLM.

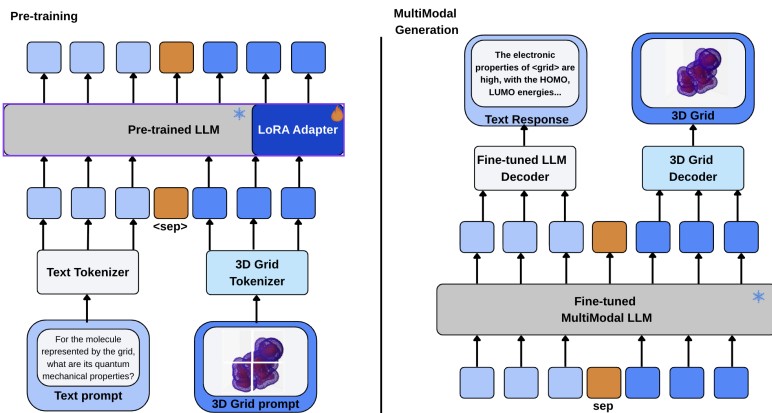

Figure 1: During training, a pre-trained large language model is equipped with LoRA adapters and fine-tuned on paired inputs consisting of 3D electron density grids—derived from either small molecules or inorganic materials—tokenized using a 3D VQGAN, and corresponding natural language prompts. Tokens from both modalities are fused at the input level, enabling early integration of spatial and textual information within a unified embedding space. After fine-tuning, the model supports both *3D-to-text* and *text-to-3D* generation.

### 2.1 GENERAL ARCHITECTURE

3D VQGAN represents 3D-grids, in addition to text, as a series of discrete tokens and takes advantage of the scaling properties of auto-regressive Transformers as in Fig. 4. Below, we define the different tokenizers used in the schema.

**3D-Grid tokenizer:** To tokenize 3D electron density grids, we employ a 3D extension of the VQGAN architecture for 3D grids introduced by Soares et al. (2025c). Given a volumetric input grid, the encoder produces a latent representation $z_e \in \mathbb{R}^{\left(\frac{H}{s}\right) \times \left(\frac{W}{s}\right) \times \left(\frac{D}{s}\right) \times k}$, where $H$, $W$, and $D$ denote the spatial dimensions, $k$ is the number of latent channels, and $s$ is the spatial downsampling factor. Each latent vector is then quantized via a learned codebook $Z$, replacing it with the nearest embedding vector.

The decoder reconstructs the original grid from the quantized latents. The model is trained to minimize a composite objective:

$$L_{\text{total}} = L_{\text{rec}} + \beta L_{\text{commit}} + \gamma L_{\text{codebook}},$$

where $L_{\text{rec}}$ denotes the reconstruction loss, $L_{\text{commit}}$ penalizes the encoder for deviation from the codebook vectors, and $L_{\text{codebook}}$ updates the codebook embeddings. To extend the original 2D

VQGAN to 3D volumetric data, we adopt architectural modifications from Ge et al. (2022); Khader et al. (2022), replacing all 2D convolutions with their 3D counterparts.

We support two types of 3D electron density grids. For small molecules, we generate re-optimized conformations using the MINDO/3 semi-empirical method as implemented in the `PySCF` electronic structure package Sun et al. (2020). Specifically, the five lowest-energy conformations are optimized, and the one with the lowest energy is selected for further calculations. This conformation is then evaluated using restricted Hartree–Fock (RHF) at the STO-3G minimal basis set level to compute the ab initio electron density. The resulting continuous charge distribution is discretized into a volumetric grid format, yielding a voxelized representation of the electron density suitable for 3D modeling.

For crystalline materials, we generate 3D electron density grids directly from Crystallographic Information Files (CIFs) as described in Hoffmann et al. (2019). Each CIF is parsed using `pymatgen` to obtain the atomic structure and lattice geometry. We then compute a continuous electron density field over a cubic grid by placing a Gaussian distribution centered at each atomic site. The contribution of each atom is weighted by its atomic number $Z$, and the total electron density at each voxel is computed as the sum of atomic contributions, assuming a fixed standard deviation $\sigma$ for all atoms. This process yields a resolution-controlled, voxelized representation of the electron density, stored as a `.npy` tensor. The approach preserves periodic boundary conditions via the `PeriodicSite` formalism and supports batch conversion across large datasets of CIF files.

**Text tokenizer:** To tokenize natural language prompts and responses, we use the tokenizer associated with a pre-trained large language model (name omitted for double-blind review). The tokenizer is extended with a special separator token `<grid>`, used to delimit different input modalities, and a vocabulary of grid tokens `<g0>` to `<g2047>` representing the VQGAN-encoded 3D volumetric grids. The tokenizer operates without modality-specific preprocessing, enabling seamless early fusion of spatial and textual information within a unified token sequence. Tokenization is performed without special tokens for answers, and truncation is applied to ensure the total sequence length does not exceed 8192 tokens. This unified vocabulary allows the model to handle multimodal inputs as flat token sequences, enabling bidirectional generation and reasoning over both 3D grids and scientific text.

**Model and Training Configuration:** We build upon the (name omitted for double-blind review) foundation model, a 8 billion parameter decoder-only causal language model pretrained on a mixture of scientific and general-domain corpora. For our task, we augment this model with lightweight Low-Rank Adaptation (LoRA) modules Hu et al. (2022) to enable efficient fine-tuning on multimodal molecular property QA pairs.

We introduce LoRA adapters with a rank $r = 8$, scaling factor $\alpha = 32$, and dropout rate of 0.05. The adapters are applied to the attention projection layers (`q_proj`, `k_proj`, `v_proj`, `o_proj`, `gate_proj`) and the input token embedding layer (`embed_tokens`).

To enable processing of volumetric 3D electron density inputs, we extend the tokenizer vocabulary with 2048 discrete grid tokens (`<g0>` to `<g2047>`) corresponding to VQGAN-encoded spatial tokens, along with a special separator token `<grid>` used to mark modality boundaries. The tokenizer operates without any modality-specific preprocessing, supporting early fusion of spatial and textual information within a flat token sequence. Maximum input length is capped at 8192 tokens.

The model is trained using the Hugging Face `Trainer` API with the following hyperparameters: 3 epochs, batch size of 1 per GPU, gradient accumulation over 1 step, and a learning rate of $6.25 \times 10^{-6}$. Optimization uses AdamW with weight decay of 0.01 and mixed-precision disabled. Training is performed on a multi-GPU setup using data parallelism with fixed seed for reproducibility.

To process the 3D modality, we encode electron density grids using a pretrained 3DGrid-VQGAN Soares et al. (2025c), resized to $128^3$ resolution and log-transformed via $\log(1 + x)$. The encoded grid tokens are prepended to the user prompt, separated by the `<sep>` token. The model is trained in an autoregressive fashion, with only the response portion supervised. Figure 2 illustrates the capabilities of the proposed multimodal foundation model trained on 3D grids.

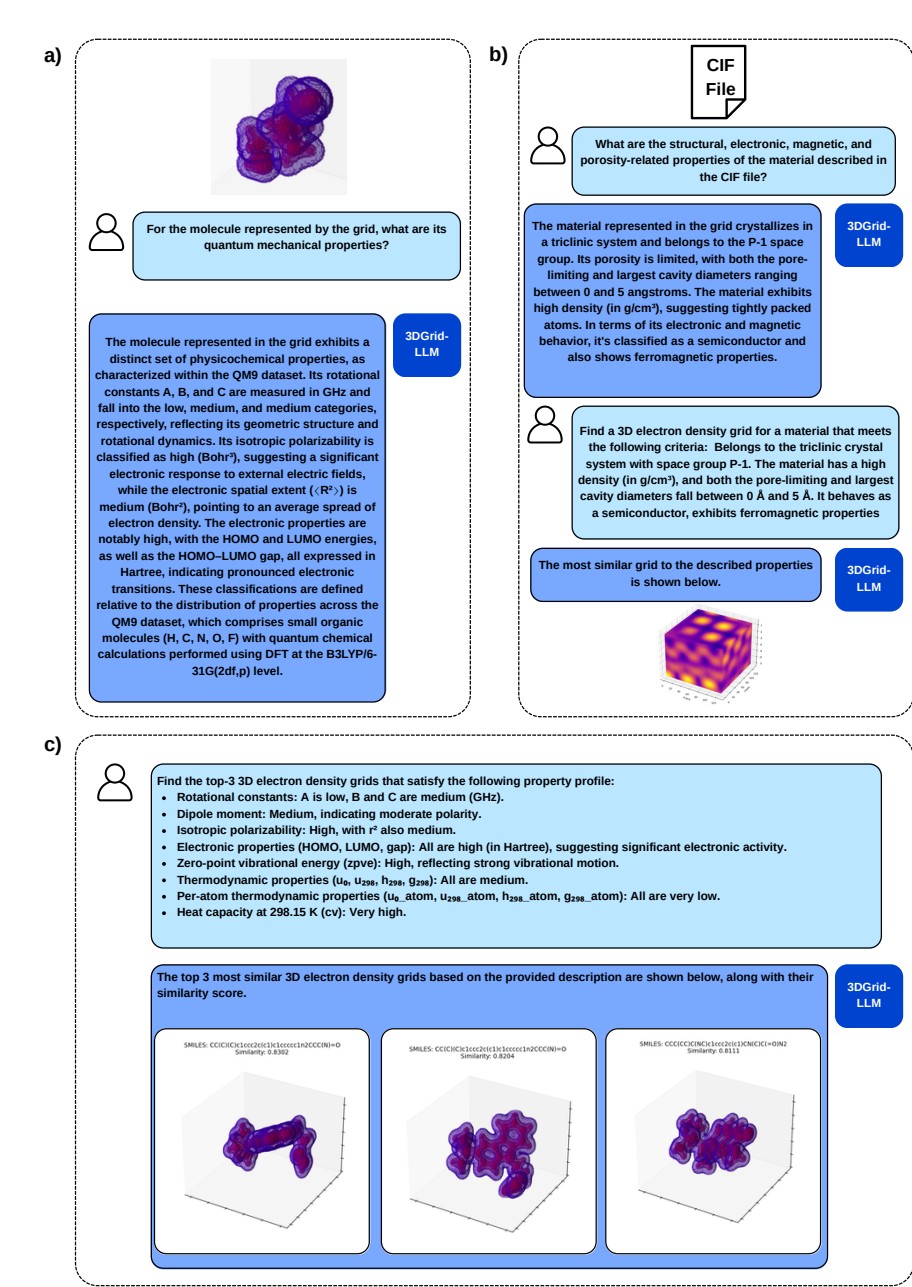

Figure 2: This figure illustrates the capabilities of 3DGrid-LLM. **a)** Given a 3D electron density grid of a molecule, the model generates structured textual descriptions of quantum mechanical properties such as rotational constants, dipole moment, polarizability, and HOMO–LUMO gap, grounded in the spatial information encoded in the grid. **b)** When provided with a CIF-derived 3D density grid, the model infers structural (e.g., crystal system, space group), electronic, magnetic, and porosity-related properties of the material in natural language. **c)** In generative-retrieval mode, the model takes a textual description of desired physicochemical or structural properties and generates discrete grid tokens, which are decoded into 3D electron density grids and compared—via learned contrastive embeddings. The top retrieved matches are presented with similarity scores.

## 2.2 PRE-TRAINING DATA

For supervised fine-tuning, we organize our dataset into three distinct categories: (i) all-properties, containing QA pairs covering multiple molecular properties; (ii) single-property, focusing on isolated property descriptions; and (iii) functional-group, which targets questions related to specific chemical substructures. These datasets are used to train the model on both 3D-grid-to-text and text-to-3D-grid tasks, enabling bidirectional understanding and generation across modalities as illustrated in Fig. 2.

The text–3D-grid data for pre-training is a combination of publicly available sources, including QM9, QMOF, and PubChem, transformed to accommodate multimodal fine-tuning. Each 3D electron density grid is resized to $128^3$ voxels and tokenized with 3DGrid-VQGAN. Across all sources, the corpus reaches 8.15 billion tokens (text + 3D-grid) spanning 12.5 million text–grid pairs. Table 1 summarizes token statistics and sample counts for each dataset.

Table 1: Token statistics for the text–3D-grid fine-tuning dataset, separated by text and grid tokens across QM9, QMOF, and PubChem.

| Dataset | Text Tokens | Grid Tokens | Total Tokens | #Samples |
|---------|-------------|-------------|--------------|----------|
| QM9 | 836M | 1.7B | 2.5B | 2.5M |
| QMOF | 9.5M | 91.8M | 101.3M | 179.2K |
| PubChem | 454M | 5.05B | 5.50B | 9.87M |
| **Total** | **1.30B** | **6.85B** | **8.15B** | **12.5M** |

## 3 EXPERIMENTS

To evaluate the proposed 3DGrid-LLM, we design a comprehensive benchmark suite spanning both Visual Question Answering (VQA) and Multimodal Retrieval tasks. Our goal is to assess the model's ability to interpret and reason over 3D electron density grids in conjunction with textual prompts, as well as its capacity to perform cross-modal alignment.

For the VQA setting, we compile a diverse set of **32 supervised tasks**, grouped into three categories based on their original dataset source:

- **PubChem**: Tasks related to molecular complexity, weight, and topological properties.
- **QM9**: Tasks derived from quantum chemistry simulations, involving rotational constants, dipole moments, electronic, and thermodynamic properties.
- **QMOF**: Tasks pertaining to structural and electronic features of crystalline materials.

The 32 VQA tasks are detailed in the Appendix, due to limit of pages.

To assess the effectiveness of our proposed 3DGrid-LLM model in generating chemically meaningful volumetric representations from property-centric prompts, we introduce a *retrieval-augmented evaluation framework* grounded in a multimodal embedding space. The pipeline, illustrated in Fig. 4, performs generation, decoding, embedding, and retrieval entirely in 3D space—bypassing reliance on molecular graph intermediates and enabling direct reasoning over electron density distributions.

Given a textual prompt describing a desired physicochemical profile, 3DGrid-LLM autoregressively generates a sequence of discrete tokens representing a latent 3D electron density grid. These tokens are decoded into a dense volumetric field ($128 \times 128 \times 128$) using a frozen 3DGrid-VQGAN decoder. The resulting grid is then passed through a contrastively trained encoder, 3DGrid-CLIP, which embeds it into a learned representation space optimized for structural and semantic alignment. We perform retrieval by comparing the embedding of the generated grid against a held-out database of experimentally or computationally derived materials, using cosine similarity to identify the top-$k$ most similar entries.

While traditional retrieval tasks in language and vision domains typically rely on ranking precision or cosine similarity, these metrics are insufficient in scientific applications where preserving *latent physical structure, property consistency, and functional diversity* is critical. To address this, we report a suite of complementary metrics that evaluate both semantic fidelity and property alignment:

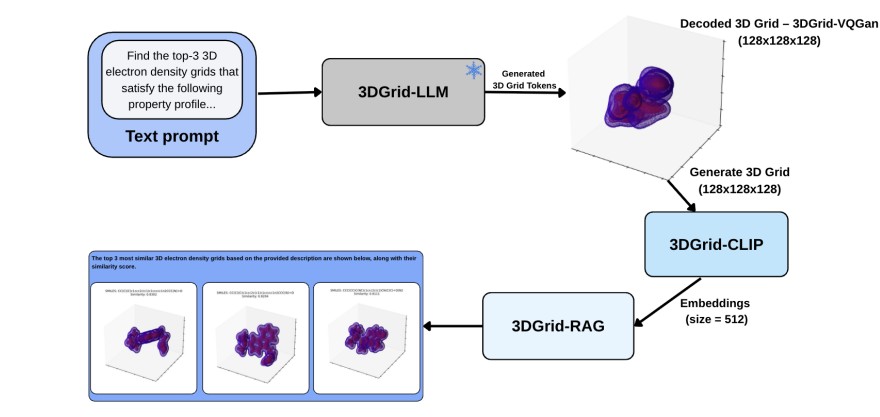

Figure 3: Schematic of the retrieval-augmented generation (RAG) pipeline. Given a textual prompt, 3DGrid-LLM generates a discrete token sequence that is decoded into a 3D grid. This grid is embedded via 3DGrid-CLIP and compared against a catalog of known materials for retrieval based on structural and semantic similarity.

- **Top-1 and Top-$k$ Similarity**: Cosine similarity between the query and retrieved embeddings.
- **Soft Recall@k**: Fraction of prompts retrieving at least one candidate from the correct property cluster.
- **Jaccard Similarity**: Overlap of discretized property bins (e.g., *low/medium/high* dipole moment).
- **BERTScore (F1)**: Semantic similarity between textual descriptions of the query and retrieved molecules.
- **Property Overlap (%)**: Percentage of shared qualitative property categories between the generated grid and the retrieved candidates.

We evaluate this framework on a benchmark set of 100 diverse textual prompts designed to elicit a range of structural and electronic characteristics. Each prompt is evaluated against a held-out catalog of 1,000 precomputed 3D electron density grids from the QMOF dataset, which provides rich property annotations and physically grounded representations of metal-organic frameworks. This setup allows us to measure how well the generated grids enable retrieval of known materials with matching physical attributes, offering a rigorous proxy for evaluating generative utility in inverse design contexts.

## 4 RESULTS

### 4.1 MULTIMODAL VISUAL QUESTION ANSWERING

Table 5 reports accuracy across 32 VQA tasks spanning general molecular, quantum-chemical, and crystallographic properties. Overall, the 3DGrid-LLM surpasses the 3DGrid-VQGAN baseline, with mean accuracy increasing from 0.5789 to 0.6766 under five-shot conditioning.

**General Molecular Properties** show an increase from 0.2123 to 0.5648 across seven tasks, with the largest gains observed in properties with near-zero baseline performance, while properties such as Topological Polar Surface Area and Complexity exhibit minimal improvement.

**Quantum Chemistry and Thermodynamic Properties** span 19 tasks and increase from 0.6436 to 0.6709. Gains are heterogeneous: structural constants and Electronic Spatial Extent improve steadily with few-shot examples, whereas properties like Heat Capacity and Enthalpy at 298.15 K show limited or variable improvement, reflecting task-dependent integration of 3D structure and textual prompts.

Table 2: Evaluation tasks for VQA and multimodal retrieval. Metric: accuracy (higher is better). **3DGrid-VQGAN** is the baseline; **3DGrid-LLM (Ours)** denotes our proposed model with/without few-shot conditioning. Per-row maxima are highlighted.

| Task | 3DGrid-VQGAN (Baseline) | 3DGrid-LLM (Ours) | | | |
|---|---|---|---|---|---|
| | | No Few-shot | Few-shot (1) | Few-shot (3) | Few-shot (5) |
| **General Molecular Properties** | | | | | |
| Exact Mass | 0.0787 | 0.2611 | 0.2632 | 0.2881 | **0.2921** |
| Monoisotopic Mass | 0.0787 | 0.3621 | 0.4567 | 0.5732 | **0.6298** |
| Molecular Weight | 0.0813 | 0.4782 | 0.4650 | 0.5972 | **0.6101** |
| Tautomer Count | 0.0004 | 0.4555 | 0.5157 | 0.5398 | **0.5432** |
| Topological Polar Surface Area | **0.5993** | 0.4912 | 0.5751 | 0.5892 | 0.5975 |
| XLogP3 | 0.0009 | 0.3231 | 0.4982 | 0.5644 | **0.5802** |
| Complexity | 0.6466 | 0.6501 | 0.6695 | 0.6602 | **0.7005** |
| *Mean (7 tasks)* | 0.2123 | 0.4330 | 0.4919 | 0.5317 | **0.5647** |
| **Quantum Chemistry and Thermodynamic Properties** | | | | | |
| Rotational Constant $A$ | 0.6216 | 0.6005 | 0.7109 | **0.7456** | 0.7339 |
| Rotational Constant $B$ | **0.7007** | 0.6792 | 0.6856 | 0.6902 | 0.6935 |
| Rotational Constant $C$ | 0.7217 | 0.7235 | 0.7654 | 0.7802 | **0.8128** |
| Dipole Moment ($\mu$) | 0.5142 | 0.6275 | 0.6445 | 0.6698 | **0.6805** |
| Isotropic Polarizability ($\alpha$) | **0.7089** | 0.6454 | 0.6688 | 0.6723 | 0.6988 |
| Electronic Spatial Extent ($r^2$) | 0.7264 | 0.7586 | 0.7702 | 0.7875 | **0.8232** |
| Zero-point Vibrational Energy (ZPVE) | 0.7375 | 0.7402 | 0.7826 | **0.8330** | 0.8301 |
| Heat Capacity ($c_v$) | **0.6887** | 0.3002 | 0.3875 | 0.4956 | 0.5235 |
| HOMO Energy | 0.5035 | 0.4972 | 0.5625 | 0.5836 | **0.6225** |
| LUMO Energy | 0.5664 | 0.5625 | 0.5782 | **0.5880** | 0.5795 |
| HOMO–LUMO Gap | **0.5614** | 0.3629 | 0.6225 | 0.6740 | **0.6892** |
| Internal Energy at 0 K ($u_0$) | **0.7257** | 0.5698 | 0.6223 | 0.6331 | 0.6009 |
| Internal Energy at 298.15 K ($u_{298}$) | **0.7231** | 0.5787 | 0.5962 | 0.6125 | 0.6856 |
| Enthalpy at 298.15 K ($h_{298}$) | **0.7231** | 0.6282 | 0.6676 | 0.6991 | 0.7127 |
| Free Energy at 298.15 K ($g_{298}$) | **0.7263** | 0.6556 | 0.6878 | 0.7032 | 0.7225 |
| Per-atom $u_0$ | 0.7219 | 0.7123 | 0.7456 | 0.7568 | **0.7809** |
| Per-atom $u_{298}$ | 0.7248 | 0.7109 | 0.7565 | 0.7589 | **0.7856** |
| Per-atom $h_{298}$ | 0.7249 | 0.6785 | 0.7092 | 0.7225 | **0.7356** |
| Per-atom $g_{298}$ | **0.7178** | 0.5674 | 0.6488 | 0.6796 | 0.6707 |
| *Mean (19 tasks)* | 0.6601 | 0.6171 | 0.6748 | 0.6992 | **0.7116** |
| **Crystallographic and Structural Properties** | | | | | |
| Crystal System | 0.5947 | 0.6032 | 0.6007 | 0.6227 | **0.6332** |
| Pore Limiting Diameter (PLD) | **0.9388** | 0.8986 | 0.9062 | 0.9065 | 0.9122 |
| Largest Cavity Diameter (LCD) | **0.9271** | 0.8134 | 0.8356 | 0.8992 | 0.9016 |
| Density | 0.8734 | 0.8189 | 0.8777 | **0.8816** | 0.8815 |
| Band Gap | 0.9558 | 0.8791 | 0.9221 | **0.9720** | 0.9684 |
| Charge | 0.5208 | 0.5765 | 0.6352 | 0.6192 | **0.6532** |
| *Mean (6 tasks)* | 0.8018 | 0.7650 | 0.7946 | 0.8169 | **0.8250** |
| **Overall mean (32 tasks)** | 0.5932 | 0.5937 | 0.6461 | 0.6748 | **0.6886** |

**Crystallographic and Structural Properties** include six tasks and increase from 0.8018 to 0.8250. Saturation is observed for Band Gap and pore size metrics, whereas Crystal System and Charge benefit more from few-shot conditioning.

Overall, few-shot conditioning selectively improves tasks with low baseline performance, while properties with strong baseline signals show diminishing returns. These results indicate that the model effectively integrates multimodal information, but the magnitude of improvement depends on both the baseline signal and the intrinsic complexity of each property.

### 4.2 ANALYSIS OF SEMANTIC GENERATION ACROSS MOLECULAR DOMAINS

To evaluate the semantic fidelity of our generative model across distinct chemical knowledge domains, we analyze BLEU, ROUGE-L, and BERTScore (F1) on structured text generation conditioned on 3D electron density grids. These metrics collectively quantify syntactic alignment (BLEU), surface-

level sequence overlap (ROUGE-L), and contextual semantic similarity (BERTScore), providing a multifaceted lens on generative quality. Table 3 illustrates the results for tested benchmarks.

Table 3: Semantic evaluation metrics across molecular datasets. BLEU captures n-gram overlap, ROUGE-L measures longest common subsequence, and BERTScore (F1) assesses contextual semantic similarity.

| Dataset | BLEU ↑ | ROUGE-L ↑ | BERTScore (F1) ↑ |
|---------|--------|-----------|------------------|
| PubChem | 0.865 | 0.918 | 0.944 |
| QM9 | 0.579 | 0.819 | 0.820 |
| QMOF | 0.782 | 0.864 | 0.878 |

As state in Table 3, 3DGrid-LLM achieves near-parity with ground-truth references in PubChem (BLEU: 0.865, ROUGE-L: 0.918, BERTScore: 0.944), underscoring its strong lexical precision and semantic alignment. This is facilitated by the categorical nature of PubChem descriptors (e.g., *logP*, *tautomer count*), which constrain linguistic variation and encourage template-consistent decoding. In contrast, performance on QM9 (BLEU: 0.579, ROUGE-L: 0.819, BERTScore: 0.820) is attenuated due to the continuous and scalar nature of quantum chemical properties (e.g., *dipole moment*, *HOMO-LUMO gap*), where the absence of standard binning leads to semantic drift and reduced surface-level overlap. Figure illustrates the answer of 3DGrid-LLM for QM9 properties.

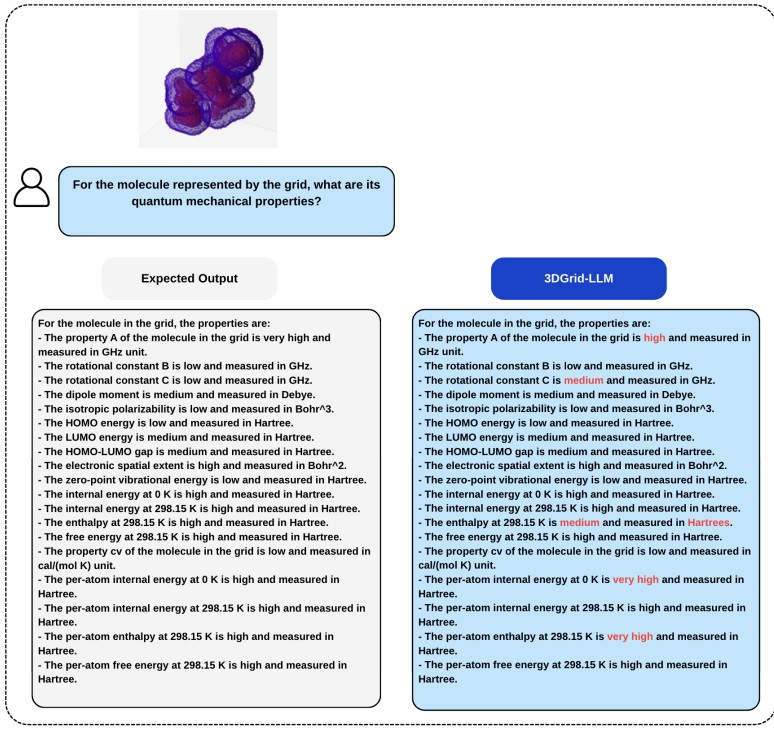

Figure 4: Example of 3DGrid-LLM answer for QM9 properties.

QMOF results (BLEU: 0.782, ROUGE-L: 0.864, BERTScore: 0.878) reflect a midpoint: the model captures structural and crystallographic features with reasonable fluency but is prone to fine-grained hallucinations, likely due to sparse and heterogeneous annotations. Overall, these findings reveal a trade-off between semantic controllability and the ontology of the property space—discrete, well-binned domains enable faithful generation, while continuous or noisy domains degrade alignment. We posit that improved grounding in such domains may require retrieval-augmented prompting or numerically constrained decoding strategies to align scalar semantics with natural language realizations.

### 4.3 RETRIEVAL-AUGMENTED 3D GRID GENERATION AND EVALUATION

We assess the generative capabilities of 3DGrid-LLM within a retrieval-augmented framework. The task consists of generating 3D electron density grids conditioned on textual property descriptions and retrieving semantically and structurally similar materials from a reference database. This setup enables a multi-modal evaluation of alignment across language, spatial representation, and functional molecular similarity.

Table 4: Retrieval performance on QMOF and QM9 datasets (Top-1 and Top-$k = 10$).

| Metric | QMOF | | QM9 | |
|---|---|---|---|---|
| | Top-1 | Top-10 | Top-1 | Top-10 |
| Cosine Similarity (Embedding Space) | 0.9794 | — | 0.9555 | 0.9340 |
| Soft Recall@10 (Cluster Match) | — | 0.980 | — | — |
| Jaccard Similarity (Discrete Properties) | 0.874 | 0.856 | 0.9181 | 0.8795 |
| BERTScore (F1) | 0.966 | 0.946 | 0.9871 | 0.9505 |
| Property Overlap (%) | 83.56 | 85.72 | 86.97 | 83.76 |

As shown in Table 4, the model achieves consistent and robust alignment across both QMOF and QM9 domains. On QMOF, generated grids yield a Top-1 cosine similarity of **0.9794**, a Jaccard similarity of **0.874**, and a BERTScore F1 of **0.966**, indicating strong agreement in both geometric and linguistic representations. Similarly, performance on QM9 reflects high fidelity, with a Top-1 cosine similarity of **0.9555**, and a Jaccard similarity of **0.9181**, validating the model's generalization across molecular complexity scales.

To further probe embedding space structure, we visualize a t-SNE projection of retrieval results on QMOF in Fig. 5. The generated query (red) and its Top-10 retrieved candidates (colored) form a dense and coherent cluster, while background entries (gray) remain distributed across the manifold. This highlights the model's precision in matching grid semantics.

Despite high accuracy, retrieved candidates display limited functional diversity, suggesting embedding collapse and reduced exploration potential. While high Top-10 Jaccard similarity (**0.8795** on QM9, **0.856** on QMOF) and property overlap indicate semantic consistency, they may mask latent redundancy. This precision-diversity trade-off is emblematic of contrastive training regimes and suggests the need for enhanced regularization. We hypothesize that diversity-aware ranking objectives, entropy-penalized decoding, or property-conditioned sampling strategies may yield broader functional coverage without sacrificing retrieval quality.

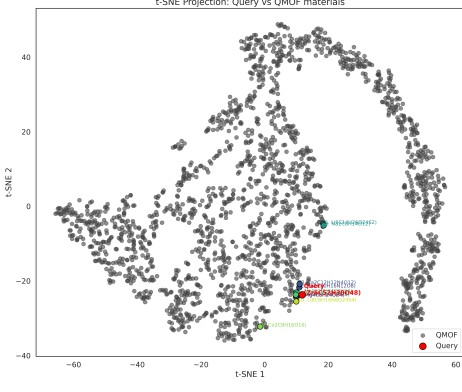

Figure 5: t-SNE projection of the 3DGrid-CLIP embedding space for a QMOF prompt. Red: generated query; Green: Top-10 retrieved; Gray: reference catalog.

## 5 CONCLUSION

We presented 3DGrid-LLM, an early-fusion multimodal foundation model that processes natural language and 3D electron density grids for bidirectional generation, reasoning, and retrieval in molecular and materials science. By extending a large decoder-only language model with discrete volumetric tokens from a 3D VQGAN, the approach captures spatial, electronic, and textual information within a unified token sequence.

3DGrid-LLM offers a scalable path to integrating physically grounded volumetric data into large language models, enabling general-purpose scientific assistants that bridge symbolic and spatial reasoning. Future work will address larger multimodal datasets, physical constraints in decoding, and new scientific modalities.

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

## A SUPPLEMENTARY MATERIALS

### A.1 LIST OF EVALUATION TASKS USED FOR VQA AND MULTIMODAL RETRIEVAL

Table 5 summarizes the 32 tasks used to benchmark 3DGrid-LLM in multimodal VQA and retrieval settings. The tasks span three domains: (i) **general molecular properties** from PubChem, covering compositional and topological descriptors such as mass, tautomer count, and lipophilicity (XLogP3); (ii) **quantum-chemical and thermodynamic properties** from QM9, including rotational constants, dipole moments, polarizability, frontier orbital energies, thermodynamic quantities, and their per-atom equivalents; and (iii) **crystallographic and structural properties** from QMOF, focusing on lattice classification, pore and cavity dimensions, density, band gap, and charge state. All tasks are formulated as classification or binning problems and evaluated uniformly using accuracy, enabling direct comparison across modalities and property types.

Table 5: List of evaluation tasks used for VQA and multimodal retrieval. All tasks are evaluated using accuracy as the metric.

| Task | Source | Evaluation Metric |
|---|---|---|
| Exact Mass | PubChem | Accuracy |
| Monoisotopic Mass | PubChem | Accuracy |
| Molecular Weight | PubChem | Accuracy |
| Tautomer Count | PubChem | Accuracy |
| Topological Polar Surface Area | PubChem | Accuracy |
| XLogP3 | PubChem | Accuracy |
| Complexity | PubChem | Accuracy |
| Rotational Constant A (A) | QM9 | Accuracy |
| Rotational Constant B (B) | QM9 | Accuracy |
| Rotational Constant C (C) | QM9 | Accuracy |
| Dipole Moment ($\mu$) | QM9 | Accuracy |
| Isotropic Polarizability ($\alpha$) | QM9 | Accuracy |
| Electronic Spatial Extent ($r^2$) | QM9 | Accuracy |
| Zero-point Vibrational Energy (ZPVE) | QM9 | Accuracy |
| Heat Capacity (cv) | QM9 | Accuracy |
| HOMO Energy | QM9 | Accuracy |
| LUMO Energy | QM9 | Accuracy |
| HOMO–LUMO Gap | QM9 | Accuracy |
| Internal Energy at 0 K ($u_0$) | QM9 | Accuracy |
| Internal Energy at 298.15 K ($u_{298}$) | QM9 | Accuracy |
| Enthalpy at 298.15 K ($h_{298}$) | QM9 | Accuracy |
| Free Energy at 298.15 K ($g_{298}$) | QM9 | Accuracy |
| Per-atom Internal Energy at 0 K ($u_0^{\mathrm{atom}}$) | QM9 | Accuracy |
| Per-atom Internal Energy at 298.15 K ($u_{298}^{\mathrm{atom}}$) | QM9 | Accuracy |
| Per-atom Enthalpy at 298.15 K ($h_{298}^{\mathrm{atom}}$) | QM9 | Accuracy |
| Per-atom Free Energy at 298.15 K ($g_{298}^{\mathrm{atom}}$) | QM9 | Accuracy |
| Crystal System | QMOF | Accuracy |
| Pore Limiting Diameter (PLD) | QMOF | Accuracy |
| Largest Cavity Diameter (LCD) | QMOF | Accuracy |
| Density | QMOF | Accuracy |
| Band Gap | QMOF | Accuracy |
| Charge | QMOF | Accuracy |

