# OpenReview forum: "Token-Level Early Fusion Model Bridging Text and 3D Electron Density Grids in Chemistry"
_ICLR.cc/2026/Conference — Submitted to ICLR 2026_

### Official Review · Reviewer_1EeF · 2025-10-29

**Soundness:** 2
**Presentation:** 2
**Contribution:** 2
**Rating:** 2
**Confidence:** 4

**Summary:**

This paper introduces 3DGrid-LLM, a multimodal foundation model designed for molecular and materials science. The method performs token-level early fusion between natural language and 3D electronic density grids.

**Strengths:**

* The paper is well-structured and clearly written, providing a concise and accessible overview of the problem and the proposed solution.
* This work demonstrates the potential of token-level early fusion strategies for advancing multimodal exploration in AI4Science.

**Weaknesses:**

* In this work, can the model ensure equivariance for the 3D electronic density grids? The performance on datasets such as QM9 is significantly poorer than that of domain-specific models—could this be due to the lack of equivariance?
* How well does the 3D electronic density grid generalize? For molecules and materials of varying sizes, how is the fidelity of the 3D electronic density representation maintained?
* The paper employs VQGAN to encode 3D volumetric grids. It would be valuable to include comparisons and discussions with VQVAE-related approaches [1-3].

[1] Van Kempen M, Kim S S, Tumescheit C, et al. Fast and accurate protein structure search with Foldseek[J]. Nature biotechnology, 2024, 42(2): 243-246.

[2] Gao K, Wang Y, Guan H, et al. Tokenizing 3d molecule structure with quantized spherical coordinates[J]. arXiv preprint arXiv:2412.01564, 2024.

[3] Li X, Wang L, Luo Y, et al. Geometry Informed Tokenization of Molecules for Language Model Generation[J]. arXiv preprint arXiv:2408.10120, 2024.

**Questions:**

See Weaknesses.

---

### Official Review · Reviewer_vdw2 · 2025-10-29

**Soundness:** 1
**Presentation:** 2
**Contribution:** 1
**Rating:** 2
**Confidence:** 4

**Summary:**

The authors propose 3DGrid-LLM, an early-fusion multimodal foundation model that bridges language modality and 3D grids modality. By tokenizing the 3D grids, the authors input them into a pretrained language model, to enable multiple downstream tasks, such as, multimodal VQA, semantic text generation, and property-aligned retrieval.

**Strengths:**

1. The paper is easy to follow and the design of the framework is generally reasonable.

2. The authors test the framework on 32 tasks, which demostrates its improved performce.

3. The topic that including into 3D grid into large language model is interesting and important.

**Weaknesses:**

1. The baselines are largely missing. The authors conduct 3 kinds of experiments, and only include one baseline in one of them. This even makes the claim 'improvements over baselines' to be 'improvements over baseline'. (Line 018).

2. The authors do not property discuss their motivation in the experimental part. The first motivation, some existing methods only employ the 1D or 2D data. The authors should at least conduct some experiments on 1D or 2D data to show the improvement from introducing 3D grid. Another way is to introduce strong related baselines. Similarly, the authors also claim their method benefits from early fusion. However, there is no such ablation study or baselines to support the claim.

3. The method is quite straightforward, and lacks of novelty. It seems the only novelty of this paper is putting text and 3D grid into the LLM together, which is clearly below the bar of ICLR.

4. The quality of presentation can be improved. The fonts in the figures are too small and resulting the SMILES in the figure is head to read.

**Questions:**

See Weaknesses.

---

### Official Review · Reviewer_VMfj · 2025-10-30

**Soundness:** 1
**Presentation:** 1
**Contribution:** 1
**Rating:** 2
**Confidence:** 5

**Summary:**

This paper introduces 3DGrid-LLM, a multimodal foundation model designed to integrate natural language with 3D electron density grids for applications in chemistry and materials science. The core idea is to achieve "early fusion" at the token level. It employs a 3D VQGAN to discretize the 3D electron density grids into volumetric tokens. These grid tokens are then combined with text tokens into a single, unified sequence that is processed by a decoder-only large language model (LLM) fine-tuned with LoRA adapters. The authors demonstrate the model's capabilities on tasks like multimodal question answering (VQA), 3D-to-text generation, and retrieval-augmented text-to-3D generation.

**Strengths:**

1. The choice of electron density grids as the 3D modality is well-justified. This representation is information-rich and captures more fine-grained electronic and spatial features than atomistic point clouds.

2. Empirical Results: On the self-defined VQA and retrieval tasks, the model shows consistent improvements over the "3DGrid-VQGAN" baseline, particularly with few-shot prompting.

**Weaknesses:**

This paper suffers from significant weaknesses that, in my opinion, make it fall below the standards of a top-tier conference like ICLR.

1. The paper's central claim to novelty is its "token-level early fusion", which it contrasts with late-fusion architectures. However, this architectural concept is not new. Ignoring both projector-based (3D-MoLM) and other token-based (3D-MolT5) state-of-the-art methods, the authors have failed to situate their work in the current research landscape.

2. Citation format error.

3. Reproductivity and ethic statement missed.

4. The method needs to be compared with more baselines, both LLM-based and conventional approaches.

**Questions:**

See weaknesses.

---

### Official Review · Reviewer_82hd · 2025-10-31

**Soundness:** 1
**Presentation:** 2
**Contribution:** 2
**Rating:** 2
**Confidence:** 4

**Summary:**

This paper introduces 3DGrid-LLM, a novel multimodal language model designed to jointly process textual input and 3D electron density grids of chemical structures. To enable the model to handle electron density grids, the authors employ a VQGAN to encode them into discrete token sequences. In addition, they propose a retrieval-augmented evaluation framework for assessing the generated electron density grids by retrieving the most similar chemical structures from a curated database of experimentally or computationally derived materials.

**Strengths:**

The paper explores an original and promising direction by using electron density grids as input data for language model training. This idea opens new avenues for integrating quantum chemistry principles into multimodal language modeling and for leveraging the abundance of existing quantum-chemical data.

Furthermore, the proposed retrieval-augmented evaluation approach is conceptually interesting. It offers a potential pathway for evaluating molecular generative models by mapping their outputs into structured chemical or material spaces, such as the domain of synthesizable molecules. This perspective could inspire future research on aligning molecular representations with experimentally verifiable chemical spaces.

**Weaknesses:**

While the paper introduces interesting ideas, its experimental validation remains limited, leaving several key questions unresolved.

1. [major] The paper does not sufficiently justify why electron density grids should be preferred over more conventional molecular representations such as 3D atomic coordinates (molecular conformations). Recent works ([a, b, c]) have already introduced spatial molecular language models operating on atomic structures using both text and specialized encoders. A comparative evaluation against such baselines is essential to substantiate the advantages of using electron density grids. Ideally, the authors should also present use cases or experiments showing scenarios where electron density grids yield clear performance or interpretability gains.
2. [major] Although the retrieval-augmented evaluation framework is novel, it inherently limits assessment to the space of existing molecules and thus cannot evaluate the model’s ability to generate truly novel structures. It is also unclear whether the proposed system can reconstruct explicit novel molecular structures from the generated electron density grids, or if it merely identifies the nearest match within a pre-existing dataset. If the latter, the practical applicability of the approach for generative tasks is significantly reduced.
3. [major] The paper mentions combining several open-access molecular and materials datasets, augmented with computed properties and textual descriptions, as well as a benchmark of 100 textual prompts designed to elicit diverse structural and electronic characteristics. However, the description of the dataset construction process is insufficient. A reproducibility statement should be included, along with high-level details about the data preparation pipeline, prompt generation examples, and data availability. These additions—possibly in the Supplementary Materials—would greatly improve transparency and replicability.
4. [minor] The section on Model and Training Configuration could be condensed or moved to the Supplementary Materials to improve readability. The main text could then focus more on experimental results and analysis, strengthening the empirical foundation of the work.

a. BindGPT: A Scalable Framework for 3D Molecular Design via Language Modeling and Reinforcement Learning, Zholus et al.

b. Towards 3D Molecule-Text Interpretation in Language Models, Li et al.

c. nach0-pc: Multi-task Language Model with Molecular Point Cloud Encoder, Kuznetsov et al.

**Questions:**

In addition to the points raised above, I have the following question:

In Table 5, all evaluation metrics are reported as “Accuracy,” yet some of the tasks involve continuous (real-valued) predictions. Could the authors clarify how “accuracy” is computed in those cases? For instance, are thresholds applied, or are these values discretized in some way?

---

### Official Review · Reviewer_huvC · 2025-11-01

**Soundness:** 2
**Presentation:** 2
**Contribution:** 2
**Rating:** 4
**Confidence:** 3

**Summary:**

The paper introduces 3DGrid-LLM, an early-fusion multimodal foundation model that unifies natural language and 3D electron-density grids for molecular and materials science. It extends a large decoder-only LLM with volumetric tokens from a 3D-VQGAN so that spatial and textual information are processed together at the token level. Trained on 12.5 M text–grid pairs (QM9, QMOF, PubChem), it supports bidirectional generation (text→3D grid and 3D→text), scientific VQA, and retrieval-augmented grid generation. Experiments show higher accuracy than a 3DGrid-VQGAN baseline across 32 tasks and strong semantic and retrieval alignment, demonstrating that early token-level fusion enables physically consistent multimodal reasoning.

**Strengths:**

- Achieves consistent performance gains and realistic bidirectional generation outputs illustrated in Figure 2.
- The architecture (3D VQGAN + LLM + LoRA adapters) and multimodal training pipeline are well-described.
- Benchmarked on diverse molecular and crystallographic datasets (PubChem, QM9, QMOF) with both VQA and retrieval metrics.

**Weaknesses:**

- The method mainly adapts existing 3D VQGAN tokenization and LoRA-based fine-tuning; architectural innovation is limited.
- No experiments isolating the impact of early fusion vs. late fusion or LoRA adaptation.

**Questions:**

- How is the 100 diverse textual prompts generated?
- The section 4.3 evaluation relies on the 3DGrid-CLIP embedding, how sensitive are the retrieval metrics to biases or saturation in that embedding space? Do the retrieved nearest neighbors correspond to physically similar electron densities, or just ones with correlated textual descriptions?
- How can the model deal with the token-level artifacts or unrealistic density patterns in 3D geometry generation?

---

### Meta-Review · Area_Chair_heRb · 2026-01-07

**Summary:**

The paper proposes 3DGrid-LLM, a multimodal foundation model designed to bridge natural language and 3D electron density grids for applications in molecular and materials science. 3DGrid-LLM is a multimodal foundation model designed to bridge the gap between natural language and three-dimensional electron density grids in the field of molecular and materials science. By using a 3D VQGAN to discretize volumetric grids into discrete tokens, the model integrates spatial and textual information through a token-level early fusion strategy within a decoder-only large language model architecture fine-tuned via LoRA. This framework enables future scientific applications.

The prevailing concerns among reviewers focus on the insufficient experimental validation, specifically the lack of comparative benchmarks against state-of-the-art models that use conventional molecular representations like atomic coordinates. Reviewers also pointed out that the architectural novelty is somewhat limited, as the combination of VQGAN and LLM adapters is a well-established pipeline, and the paper fails to provide ablation studies to prove that early fusion is superior to late-fusion alternatives. Furthermore, there are significant technical doubts regarding the model's ability to maintain physical equivariance and consistency, alongside a call for greater transparency regarding the dataset construction process and overall reproducibility of the results.

The authors also didn't provide any response.

**Reviewer Concerns:**

see metareview

**Reviewer Scores:**

see metareview

---

### Decision · Program_Chairs · 2026-01-26

Reject